# Freshwater Clam Extract Attenuates Indomethacin-Induced Gastric Damage In Vitro and In Vivo

**DOI:** 10.3390/foods12010156

**Published:** 2022-12-28

**Authors:** Fuad Sauqi Isnain, Nai-Chen Liao, Hui-Yun Tsai, Yu-Jie Zhao, Chien-Hua Huang, Jue-Liang Hsu, Agustin Krisna Wardani, Yu-Kuo Chen

**Affiliations:** 1Department of Agricultural Product Technology, Faculty of Agricultural Technology, Brawijaya University, Malang 65145, Indonesia; 2Department of Biological Science and Technology, National Pingtung University of Science and Technology, Pingtung 91201, Taiwan; 3Department of Food Technology, Faculty of Engineering, Bumigora University, Mataram 83127, Indonesia; 4Department of Food Science, National Pingtung University of Science and Technology, Pingtung 91201, Taiwan; 5Department of Nutrition and Health Science, Fooyin University, Kaohsiung 83102, Taiwan; 6Aging and Disease Prevention Research Center, Fooyin University, Kaohsiung 83102, Taiwan

**Keywords:** *Corbicula fluminea*, NSAIDs, gastroprotection, inflammation, antioxidant, food nutrition improvement

## Abstract

Contemporary pharmacological studies have reported that freshwater clam (*Corbicula fluminea*) can provide a broad spectrum of bioactivities, including antioxidant, anticancer, antihypertensive, hepatoprotective, and hypocholesterolemic effects. The aim of this study was to evaluate the gastroprotective effects of water extract of freshwater clam (WEC) on indomethacin (IND)-induced gastric mucosal cell damage in vitro and gastric ulcer in vivo. The cell viability of rat gastric mucosa RGM-1 cells was markedly decreased by 0.8 mM of IND treatment, and pre-treated with various concentration of WEC significantly restored IND-induced cell damage in a dose-dependent manner. WEC also significantly attenuated the elevated reactive oxygen species (ROS) levels, inducible nitric oxide synthase (iNOS) and cyclooxygenase-2 (COX-2) expression, and nuclear factor-κB (NF-κB) p65 nuclear translocation induced by IND. In the in vivo study, IND caused severe gastric ulcer in Wistar rats, while WEC pretreatment effectively reduced the ulcer area and edema in the submucosa. We found that WEC significantly restored glutathione (GSH) content in gastric mucosa in a dose-dependent manner (*p* < 0.05). The reduction of prostaglandin E_2_ (PGE_2_) caused by IND was also improved with higher doses of WEC administration. Moreover, the overexpression of COX-2, iNOS, and tumor necrosis factor-α (TNF-α) proteins in gastric mucosa was downregulated by administration of WEC. Consequently, WEC can be used as a potential nutritional supplement to improve NSAIDs-caused gastric mucosal lesions.

## 1. Introduction

Under the condition of disordered eating and anxious working rhythm, modern people often suffer from gastrointestinal-related disorders, such as gastric ulcer, gastroesophageal reflux, indigestion, etc. Gastric ulcer is one of the most common gastrointestinal diseases, affecting at least 10% of the world’s population [1]. Factors that cause gastric ulcer include genetic predispositions, *Helicobacter pylori* infection, stress, smoking, nutritional deficiencies, alcohol consumption, excessive gastric acid secretion, and overuse of nonsteroidal anti-inflammatory drugs (NSAIDs) [2,3]. Evidence from several reports indicates that gastric cancer and gastric ulcer share many of the same etiologies, and gastric ulcer patients are a high-risk group for gastric cancer [4,5]. It can be seen that how to treat gastric ulcer and prevent its occurrence is indeed an extremely important issue for people in today’s society.

NSAIDs are widely used in the treatment of fever, pain relief, and anti-inflammation. However, overuse of this class of drugs causes certain side effects, especially in the digestive tract, such as gastric mucosal erosions, ulcers, bleeding, and perforation [6]. Numerous studies have demonstrated that these gastrointestinal symptoms caused by NSAIDs are related to the reduction of prostaglandin synthesis and the inhibition of epithelial cell regeneration [7,8,9]. The pathological mechanism of gastric mucosal injury or hemorrhage caused by NSAIDs involve inhibition of activities of cyclooxygenase-1 (COX-1) and COX-2 in gastric mucosa and direct toxicity to gastric epithelial cells [10]. Inhibition of COX-1 activity results in decreased gastric mucosal blood flow and decreased gastric mucus and bicarbonate secretion, all of which reduce the defense system of gastric mucosal cells. The reduction of COX-1 and COX-2 activities inhibits the synthesis of prostaglandins, thereby reducing the integrity of the gastric mucosa, causing a large number of neutrophils to accumulate in the gastric mucosa, increasing oxidative stress and inflammatory responses and causing gastric ulcers [11]. Therefore, the regulation of COX/prostaglandin pathway is an important factor that must be considered for the prevention or treatment of gastric mucosal damage caused by NSAIDs.

Indomethacin (IND), a kind of NSAIDs, is commonly used to reduce several clinical symptoms, such as fever, pain, stiffness, swelling, etc. However, excess consumption of IND causes damage to the gastric mucosa. The main reason is that IND inhibits the production of prostaglandins, which protect the mucosal cells of the digestive tract by maintaining gastric mucosal blood flow and increasing the secretion of mucus and bicarbonate [12]. Compared with other NSAIDs, IND induces more severe gastric mucosal damage in rats. For this reason, IND is frequently used in many in vitro and in vivo studies to induce gastric mucosal cell damage to explore the gastroprotective effects of test samples [13,14,15,16]. In the IND-induced gastric mucosal injury model, the increase of reactive oxygen species (ROS) is also an important role. Previous studies have shown that IND-induced gastric mucosal damage is associated with the disruption of intracellular antioxidant defenses as glutathione (GSH) and the accumulation of ROS which cause lipid peroxidation (LPO) in mucosal cells, whereas chronic oxidative stress in mucosal tissue resulted in gastric ulcer [17]. In addition, IND administration caused nuclear translocation of nuclear factor-κB (NF-κB), which promoted the expression of COX-2 and inducible nitric oxide synthase (iNOS) to increase the inflammatory response [18,19]. This could also explain the increased expression of COX-2 in the IND-treated gastric ulcer area in animal models.

Freshwater clam (*Corbicula fluminea*), also known as Asiatic clam or river clam, native to East and South Asia, is widely distributed in the bottom sediments of lakes, rivers, and ponds in fresh or brackish water. Previous studies have reported that freshwater clam has several health benefits, such as the regulation of blood lipids, the improvement of fatty liver, and anticancer, hepatoprotective, and antioxidative properties [20,21,22,23,24,25]. Accordingly, freshwater clam is currently widely used as supplements and health food in East Asia, such as Japan, Taiwan, and China. Aqueous extracts from freshwater clam contain amino acids (or peptides), phenolic compounds and steroid components that are thought to show antioxidant activity [26]. In the present study, IND-induced gastric mucosa injury in vitro and in vivo models were used to explore the protective activities of water extract of freshwater clam (WEC) and to further elucidate the underlying mechanism.

## 2. Materials and Methods

### 2.1. Chemical Reagents

Dulbecco’s Modified Eagle Medium/Nutrient Mixture F-12 (DMEM/F-12), fetal bovine serum (FBS), glutaMAX, and penicillin-streptomycin (PS) were obtained from Gibco (Grand Island, NY, USA). Coomassie protein assay reagent kit and Minute™ Cytoplasmic and Nuclear Extraction Kit were purchased from Thermo Fisher Scientific (Waltham, MA, USA). The COX-1, COX-2, and NF-κB p65 antibodies were obtained from Cell Signaling Technology (Danvers, MA, USA). The iNOS and TNF-α antibodies were provided by Proteintech (Rosemont, IL, USA). Amersham^TM^ ECL Select^TM^ Western Blotting Detection Reagent and Amersham^TM^ Hybond P Western blotting membranes (PVDF) were purchased from Merck (Darmstadt, Germany). The β-Actin, RIPA lysis buffer, and secondary antibodies were obtained from Merck Millipore (Burlington, MA, USA). Prostaglandin E_2_ (PGE_2_) Express ELISA Kit (500141) was provided by Cayman Chemical (Ann Arbor, MI, USA). Reduced Glutathione (GSH) Colorimetric Assay Kit was obtained from Elabscience Biotechnology (Houston, TX, USA). Protease inhibitor cocktail was proved by Fivephoton Biochemicals (San Diego, CA, USA). 2′,7′-Dichlorofluorescin diacetate (DCFH-DA), indomethacin (IND), sodiumdodecyl sulfate (SDS), trypsin-EDTA, 3-(4,5-dimethyl-2-thiazolyl)-2,5-diphenyl-2H-tetrazolium bromide (MTT), and all other chemicals were purchased from Sigma-Aldrich (St. Louis, MO, USA).

### 2.2. Preparation of Water Extract of Freshwater Clam (WEC) and Analysis of Its Amino Acid Composition

The dried powder made from freshwater clam meat was kindly provided by Li Chuan Aquafarm Co., Ltd. (Hualien, Taiwan), and the water extract of freshwater clam (WEC) was prepared according to the method in the previous study [27] with slight modifications. One kg of clam powder was weighed and then mixed with distilled water at a ratio of 1:20 (*w*/*v*) and stirred at room temperature for 24 h. The mixture was further filtered, and the filtrate was concentrated and lyophilized to obtain WEC. The composition of amino acids of WEC and its hydrolysate, which was obtained from acid-hydrolysis with 6 M HC1 at 110 °C for 24 h, were analyzed by an automatic amino acid analyzer (L-8900 AAA, Hitachi High Tech Analytical Science Ltd., Tokyo, Japan).

### 2.3. Cell Culture and Cell Viability Analysis

The rat gastric mucosal epithelial RGM-1 cells were kindly provided by Prof. Hirofumi Matsui (University of Tsukuba, Japan) and cultured in DMEM/F12 medium with 20% FBS, 1% PS, and 1% GlutaMAX. The cells were incubated at 37 °C in a humidified atmosphere of 5% CO_2_. The MTT assay was performed according to the method in the previous study [28] with slightly modifications to measure the cell viability of RGM-1 cells treated with various concentrations of WEC for 2 h or subsequently exposed to 0.8 mM IND for 6 h.

### 2.4. Reactive Oxygen Species (ROS) Production Analysis

RGM-1 cells were seeded in a 10-cm petri dish (1 × 10^6^ cells/dish) and the cells were maintained in DMEM/F12 medium with 20% FBS, 1% PS, and 1% GlutaMAX overnight at 37 °C. Replaced the culture medium with or without 62.5, 125, and 250 μg/mL WCE containing medium and incubated for 2 h, and then the cells were treated with 0.8 mM IND for 6 h. The IND-containing medium was removed and then changed to medium with 10 μM DCFH-DA working solution and incubated at 37 °C for 30 min in the dark. This was washed twice with 4 mL phosphate-buffered saline (PBS) after removing the DCFH-DA working solution. Cells were detached and centrifuged at 300× *g* for 3 min, and then resuspended in 1 mL ice-cold PBS. Finally, the fluorescence intensity was measured using a fluorescence spectrometer (Hitachi F-2700, Hitachi High Tech Analytical Science Ltd., Tokyo, Japan) at an excitation wavelength of 485 nm and an emission wavelength of 530 nm.

### 2.5. Animal Experiment

Six-week-old male Wistar rats were obtained from BioLASCO (Taipei, Taiwan) and housed in an air-conditioned room with 12 h light/dark cycle, temperature condition of 23 ± 2 °C, and relative humidity of 50–60%. The rats had free access to food and water. All animal studies were conducted under protocol number NPUST-107-069 approved by the National Pingtung University of Science and Technology, Institutional Animal Care and Use Committee. After 1 week of housing, the rats were randomly divided into five groups (n = 6), which were (1) Control group; (2) IND group; (3) 100 mg/kg WEC group; (4) 200 mg/kg WEC group; (5) 500 mg/kg WEC group. The Wistar rats in groups 3–5 were orally administered with indicated dose of WEC for 7 consecutive days, whereas those in groups 1–2 were given saline as vehicle. The rats were fasted overnight on day 6 of the experiment, followed by a single dose of IND (80 mg/kg) to rats in groups 2–5, and sacrificed using CO_2_ euthanasia 6 h later.

### 2.6. Ulcer Area and Histopathological Assessment

After the rats were sacrificed, the gastric tissues were collected and rinsed with ice-cold saline. The mucosa of the entire stomach was photographed and the damage was observed, and the tissue was then divided into two parts. One part was fixed in 10% formalin, and the other part was frozen in liquid nitrogen and then stored at −80 °C. The photos obtained were analyzed and the ulcer area was measured by the ImageJ software (National Institutes of Health, Bethesda, MD, USA). The preventive index of WEC was calculated according to the below formula reported by Robert et al. [29]:Preventive index (%) = [(Ulcer area_IND group_ − Ulcer area_WEC group_)/Ulcer area_IND group_] × 100%(1)

The histopathological assessment was conducted according to the method in the previous study [30] with certain modifications. Gastric tissues were formalin-fixed and paraffin-embedded. Before stained by hematoxylin and eosin (H&E), 5 μm thickness of section was taken and placed onto glass slides and deparaffinized. Furthermore, all sections were photographed and examined via the light microscope.

### 2.7. Analysis of the Contents of Glutathione (GSH) and Prostaglandin E_2_ (PGE_2_) in Gastric Mucosa

To analyze the contents of GSH and PGE_2_ in gastric mucosa, gastric tissues were homogenized and then centrifuged to obtain the supernatant. Then, the contents of GSH and PGE_2_ were determined using commercial assay kits (Elabscience Biotechnology, Houston, TX, USA) following the manufacturer’s guidance.

### 2.8. Protein Expression Analysis

RGM-1 cells and rat gastric mucosal tissues were homogenized with RIPA lysis buffer containing protease inhibitors to prepare the protein solutions, and Western blot was performed to analyze the protein expression as previously described [31]. Briefly, proteins were transferred from the gels onto polyvinylidene difluoride (PVDF) membranes after separated by sodium dodecyl sulfate polyacrylamide gel electrophoresis (SDS-PAGE). The PVDF membranes were then blocked with PBS containing 5% non-fat milk and 0.1% tween 20, followed by incubations with a series of primary anti-bodies against COX-1, COX-2, iNOS, NF-κB and TNF-α (1:1000) at 4 °C overnight. After washing with PBST (PBS with 0.1% tween 20), the membranes were incubated with horseradish peroxidase-labeled secondary antibody (1:5000 dilution, General Electric, Boston, MA, USA) at room temperature for 1 h. Next, the signals of the bands were developed using Amersham^TM^ ECL Select^TM^ Western Blotting Detection Reagent. The fluorescent bands were then detected by Luminescence Image system (M2-8068, Hansor, Taichung, Taiwan), and their densitometry intensities were quantified using ImageJ software.

### 2.9. Statistical Analysis

The data of the experimental results are represented as means ± standard deviation (SD). The analytical data were statistically analyzed using Statistical Product and Service Solutions 12.0 (SPSS 12.0) software (SPSS Inc., Chicago, IL, USA). Comparisons of statistical significance between groups were determined by one-way analysis of variance (ANOVA) with a Duncan’s test. A *p*-value below 0.05 was considered statistically significant.

## 3. Results and Discussion

### 3.1. Amino Acid Composition of WEC and Its Hydrolysate

Studies have shown that clam is rich in free amino acids and peptides, which should be responsible for the health benefits of freshwater clam [32,33,34]. In addition, amino acids and peptides were reported to possess protective effects on digestive tracts [35,36]. For that reason, we analyzed the amino acid composition of WEC, including free amino acids and those from hydrolysate. Table 1 shows the number of amino acids in WEC and hydrolyzed WEC by total amino acid analysis. It can be found that in the results of free amino acids, WEC has high content in certain essential amino acids, such as Thr, Val, Met, Phe, Ile, Leu, and Lys. Among them, the branched-chain amino acids, Leu, Ile, and Val, stand out. Although the Glu was the highest in the WEC hydrolyzate, those essential amino acids measured in the free amino acids also had a certain content in the WEC hydrolyzate. A study reported by Gershon et al. [37] indicated that essential amino acids play vital roles in the regulation of mucosal proliferation and inflammatory response mitigation in the gastrointestinal inflammation. Moreover, branched-chain amino acids were found to improve the intestinal defense system via conserving villous morphology and enhancing immunoglobulin generation [38]. In general, peptides containing branched-chain amino acids belonging to essential amino acids show high antioxidant activity due to their high hydrophobicity [39]. According to the above results, WEC is rich in amino acids and peptides, suggesting that it may play a role in relieving inflammation or damage of the gastrointestinal tract.

### 3.2. Effects of WEC on Cell Viability and IND-Induced Cytotoxicity in Rat Gastric Mucosa RGM-1 Cells

RGM-1 cells were first treated with 31.25, 62.50, 125, and 250 μg/mL of WCE for 2 h, and then the cell viability was analyzed by MTT assay. As shown in Figure 1A, WEC treatment at these concentration ranges had no cytotoxic effect on RGM1 cells with cell viability of 114 ± 12%, 101 ± 4%, 104 ± 2%, 99 ± 15%, and 100 ± 5%, respectively. Therefore, 250 μg/mL was subsequently used as the maximum concentration to evaluate whether WEC possesses protective activity against IND-induced RGM-1 cell damage. The results showed that pretreatment of WEC for 2 h dose-dependently restored the cell viability of RGM-1 cells, which were reduced by exposure to 0.8 mM IND for 6 h (Figure 1B). It is suggested that WEC exhibited a preventive effect on gastric mucosal cell injury caused by IND. It is suggested that one of the possible reasons why WEC exhibited the ability to protect gastric mucosal cells is that it is rich in several free amino acids or peptides, especially essential amino acids. Studies indicated that essential amino acids showed ameliorated effects on inflammation or injury in the gastrointestinal mucosa [40,41]. Urushidani et al. [35] reported that several amino acids, including essential amino acids, improved IND-caused gastric ulcers in rats. Furthermore, Abu Bakar et al. [36] found that polypeptide K, a kind of peptides identified in the seeds of *Momordica charantia*, demonstrated gastroprotective effects on IND-induced gastric ulcer models.

### 3.3. Effects of WEC on IND-Induced Oxidative Stress and Associated Protein Levels

IND causes gastric ulcers through various processes, including inhibition of prostaglandin synthesis, initiation of lipid peroxidation, accumulation of neutrophils, and induction of oxidative stress [42,43]. We further explored whether the protective effect of WEC was related to its ability to alleviate the reactive oxygen species (ROS) production caused by IND. From the results shown in Figure 2A, it can be seen that the amount of ROS produced was significantly increased after 6 h of incubation of RGM-1 cells in medium containing 0.8 mM IND (*p* < 0.05). Our result is similar to the previous research, which indicated that IND treatment resulted in the release of ROS via interference of mitochondrial membrane potential in RGM-1 cells [44]. Pretreatment of WEC showed a significant and dose-dependent decrease in the percentage of ROS production (*p* < 0.001), representing that WEC could reduce the oxidative stress induced by IND in RGM-1 cells. Despite IND inhibiting the activities of COX-1 and COX-2 in gastric mucosa cells, the inflammatory reaction was enhanced by stimulating the expression of COX-2 and iNOS, which was promoted via nuclear translocation of NF-κB in IND-administered cells [18,19]. For this reason, we analyzed the protein levels of COX-2 and iNOS, as well as nuclear translocation of NF-κB in IND-treated RGM-1 cells. The results showed that IND significantly increased the protein levels of COX-2, iNOS, and intranuclear NF-κB, while pretreated with WEC decreased the expression of these inflammatory response-related proteins in a dose-dependent manner (Figure 2B–D). Fatty acids in freshwater clam extract were found to suppress NF-κB pathway and reduce the COX-2 and iNOS expression, thereby constructively regulate the lipopolysaccharide-induced inflammation [45]. Moreover, essential amino acids in WEC, such as Phe, Met, Val, Ile, and Lys, were considered to have notable effects on digestive tract inflammation [46]. Consequently, our results demonstrated that the inhibition of NF-κB pathway, which, at least in part, is involved in the preventive effects of WEC against IND-induced injury in RGM-1 cells via suppression of COX-2 and iNOS expression.

### 3.4. Effects of WEC on IND-Induced Gastric Ulcers in Wistar Rats

We found that WEC has considerable preventive effects on gastric cell damage caused by IND in vitro; therefore, we further assessed whether WEC exerts a similar efficiency in vivo. Male Wistar rats were administered orally with 100, 200, and 500 mg/kg of WEC for 7 consecutive days. At the end of the experiment, 2 h after WEC administration, rats were treated with 80 mg/kg of IND and sacrificed 6 h later. The gastric tissue of rats was collected to examine the ulcer area and the preventive index of WEC. As can be seen in Figure 3, there were no ulcers observed on the gross appearance of gastric mucosa in the control group, while in the vehicle group, treatment of IND resulted in severe hemorrhagic streaks and ulcers. Our results are consistent with the study conducted by Kwon et al. [47] that IND treatment led to gastric mucosal lesions, such as bleeding and other ulcer symptoms. WEC administration markedly decreased the number of hemorrhagic streaks and improved the status of ulcers, with the best remission in the 500 mg/kg WEC group. Table 2 shows the ulcer area and preventive index of WEC. Similar to the results observed from the appearance of gastric tissue, the ulcer area in the IND group was as high as 74.9 ± 25.5 mm^2^, whereas administration of WEC dose-dependently reduced the ulcer area of the gastric mucosa in rats. The ulcer areas of the groups given 100, 200, and 500 mg/kg of WEC were 47 ± 14, 37 ± 9, and 20 ± 6 mm^2^, respectively. Based on the above results, WEC at 500 mg/kg had the highest preventive index calculated from the results of ulcer area. From the results of the free amino acid content of WEC (Table 1), it can be seen that WEC contains a significant amount of essential amino acids, and a previous study has shown that essential amino acids possess protective activity against IDN-caused gastric ulcer [35]. The histopathological analysis of gastric section was performed by H&E staining. As shown in Figure 4, severe gastric mucosal damage, decreased periulcerative mucosal thicknesses, and edema of submucosa were found in rats of the IND group compared to those of the control group. According to the previous study, the treatment of IND caused the destruction of the glandular structure of the gastric mucosa, resulting in the damage of the surface epithelium and significant necrotic lesions. In addition, IND also caused the loss of cell epithelium or the occurrence of inflammatory responses, and edema, inflammatory cell infiltration, release of reactive oxygen species metabolites, and cell membrane peroxidation appear at the site of inflammation [48]. Upon administration of various doses of WEC, rats showed obvious alleviative effects on the damage of mucosa and the edema of submucosa of the stomach, and WEC at 500 mg/kg showed the highest improvement efficiency. A dose of 100–500 mg/kg for rats is equivalent to the dose of about 16–80 mg/kg for humans, and the daily intake of adults is about 1.0–4.8 g based on an adult body weight of 60 kg. Such dosage ranges should be possible and feasible for adults. A number of studies reported that the essential amino acids and peptides exert anti-inflammatory effects through their own antioxidant activity or by enhancing the antioxidant system of gastric mucosal cells [26,34,39,49,50]. In the present study, the WEC we used is rich in essential amino acids (or peptides), and administration of WEC effectively protected gastric mucosal damage induced by IND via anti-inflammatory activity.

### 3.5. Effects of WEC on the Contents of GSH and PGE2 and Protein Levels of COX-1 and COX-2 of Gastric Mucosa in IND-Treated Wistar Rats

ROS generation and oxidative stress play important roles in IND-induced pathogenesis in stomach [51]. GSH, composed of Glu, Cys, and Gly, is a most abundant low-molecular weight thiol in mammalian cells. Antioxidant defense systems in organisms such as GSH can reduce or prevent gastric mucosal damage caused by ROS. Figure 5A shows that IND significantly reduced GSH content in the gastric mucosa of rats compared with the control group (*p* < 0.05), while preadministration of WEC significantly restored the decreased GSH content of rats (*p* < 0.05 and *p* < 0.01). WEC contains high amounts of free amino acids as well as those from hydrolysates, which is suggested to be crucial for maintaining a regular GSH level in organisms. For example, an adequate supply of sulfur-containing amino acids such as Met and Cys is essential for synthesis of GSH [52]. Furthermore, the most abundant amino acid in WEC hydrolysates is Glu, which is one of the constituents of GSH. Our results demonstrated that WEC dramatically recovered the decrease in GSH content caused by IND, which is beneficial for protecting gastric mucosal cells from the harmful effects of ROS. Similar results can also be seen in the article published by Chi et al., where WEC effectively restored the GSH content in injured primary cultured rat hepatocytes by CCl_4_ [53]. In the present study, we also found that IND decreased the PGE_2_ content and COX-1 expression in the gastric mucosal tissue (Figure 5B,C). Prostaglandins are key factors in activating the healing mechanism in ulcers of the digestive tract. Most of prostaglandins are synthesized by gastric mucosal cells through COX-1, and PGE_2_ is the most important prostaglandin in the human gastrointestinal tract. IND is thought to cause ulcers mainly by inhibiting the expression of COX-1 and thereby reducing the synthesis of PGE_2_ [54]. In the group administered with WEC, the content of PGE_2_ showed an upward trend, and there was a significant difference between the highest dose group (500 mg/kg WEC) and the IND group (*p* < 0.05). In addition, WEC preadministration at 200 and 300 mg/kg also significantly upregulated the expression of COX-1 (*p* < 0.05). The above results indicate that preadministration of PG improved the decline of COX-1 expression and the inhibition of PGE_2_ synthesis induced by IND, thereby improving the mucosal regeneration ability and protecting the gastric mucosa. Figure 5D shows that the COX-2 expression of gastric mucosal tissue in rats treated with IND. Consistent with the results we found in in vitro study, IND significantly upregulated the protein expression of COX-2 in gastric mucosa, while administration of WEC decreased the expression of COX-2, with the most significant effect at the highest dose (500 mg/kg WEC; *p* < 0.01). It is suggested that the expression of COX-2 increased when the tissues were under pathological states such as inflammation and damage [55]. Our results show that administration of WEC significantly reduced the expression of COX-2 and improved the inflammation and injury of gastric mucosa in rats.

### 3.6. Effects of WEC on Protein Levels of TNF-α and iNOS of Gastric Mucosa in IND-Treated Wistar Rats

The effects of WEC on gastric mucosal TNF-α and iNOS expression in IND-treated rats were further evaluated by Western blot. As is shown in the Figure 6, treatment of IND significantly increased the expression of TNF-α and iNOS compared with the control group, while preadministration of WCE significantly suppressed their expression of gastric mucosa in rats. TNF-α is an inflammatory cytokine mainly produced by activated macrophages and plays a vital role in promoting inflammation. A study conducted by Huang et al. showed that freshwater clam extract exerted anti-inflammatory and anti-TNF-α activities [56]. In addition, several studies reported that essential amino acids showed beneficial effects on gastrointestinal inflammation by mediating TNF-α [46]. Phe was reported to decrease TNF-α generation to improve inflammatory bowel disease [41]. As we mentioned above, the presence of Met is necessary to maintain stable synthesis of GSH, and the maintenance of digestive tract homeostasis by GSH may be associated with the suppression of TNF-α-mediated enhancement of paracellular permeability by Met [57]. Poly-L-lysine, a kind of polymer consisting of Lys, inhibited generation of interleukin-8 (IL-8), a pro-inflammatory chemokine, in the intestinal epithelial cells stimulated by TNF-α [58]. According to the above studies, WEC contains a considerable content of essential amino acids, so this may be one of the main reasons that WEC showed a regulation effect on the expression of TNF-α. Furthermore, WEC significantly downregulated the IND-induced increase in iNOS expression in our in vitro and in vivo findings. Activation of iNOS and COX-2 is involved in the NF-κB pathway, which leads to the production of NO and pro-inflammatory cytokines such as IL-6 and TNF-α [59]. It is reported that the fatty acids in freshwater clam extract constructively inhibited the NF-κB pathway and downregulated the expression of iNOS and COX-2, thereby suppressing the development of inflammation [45]. Therefore, our findings indicate that the regulation of NF-κB-iNOS-COX-2-TNF-α pathway plays a vital role in protective effects of WEC on IND-induced gastric mucosal injury in rats.

## 4. Conclusions

In the present study, we found that WEC showed significant protective effects against IND-induced gastric mucosal damage in vitro and in vivo. The regulation of NF-κB-iNOS-COX-2-TNF-α pathway is highly associated with gastroprotective activity of WEC. The beneficial effects of WEC could result from the presence of essential amino acids and oligo-peptides through an anti-inflammatory potential; however, the gastric mucosal protective efficacy of these essential amino acids and oligo-peptides must be investigated in the future to elucidate their role in WEC. In conclusion, WEC has the potential to improve gastric mucosal injury caused by NSAIDs, and it can be developed as a relevant nutritional supplement in the future.

## Figures and Tables

**Figure 1 foods-12-00156-f001:**
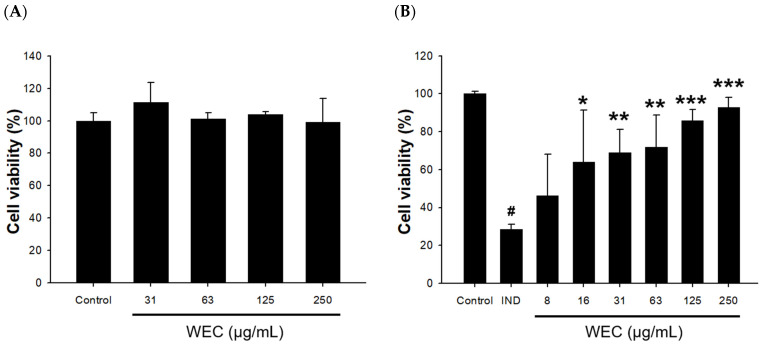
Effects of WEC on cell viability in rat gastric mucosa RGM-1 cells. (**A**) Cells were treated with different concentrations of WEC for 2 h. (**B**) Cells were treated with 0.8 mM of IND for 6 h after an 2 h pretreatment of various concentrations of WEC. Data are represented as means ± SD (n = 3). # *p* < 0.05 compared with the control group. * *p* < 0.05, ** *p* < 0.01, *** *p* < 0.001 compared with the IND-treated group.

**Figure 2 foods-12-00156-f002:**
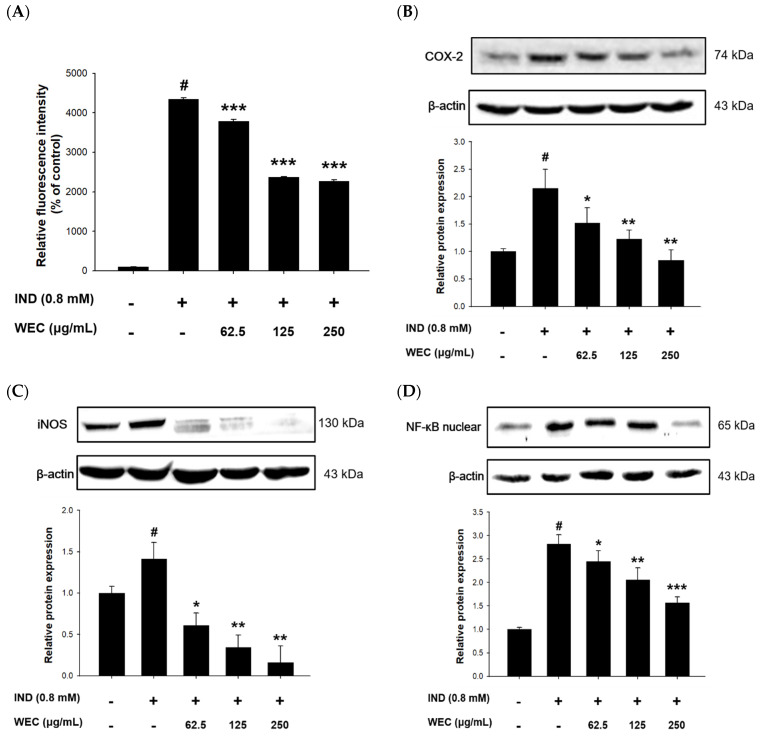
Effects of WEC on ROS generation and related protein levels in rat gastric mucosa RGM-1 cells. Cells were treated with 0.8 mM of IND for 6 h after 2 h pretreatment of various concentrations of WEC. (**A**) ROS generation percentage were determined and calculated by flow cytometry. The protein expression of (**B**) COX-2, (**C**) iNOS, and (**D**) intranuclear NF-κB was analyzed by Western blot. Data are represented as means ± SD (n = 3). # *p* < 0.05 compared with the control group. * *p* < 0.05, ** *p* < 0.01, *** *p* < 0.001 compared with the IND-treated group. + and − indicate with and without treatment, respectively.

**Figure 3 foods-12-00156-f003:**
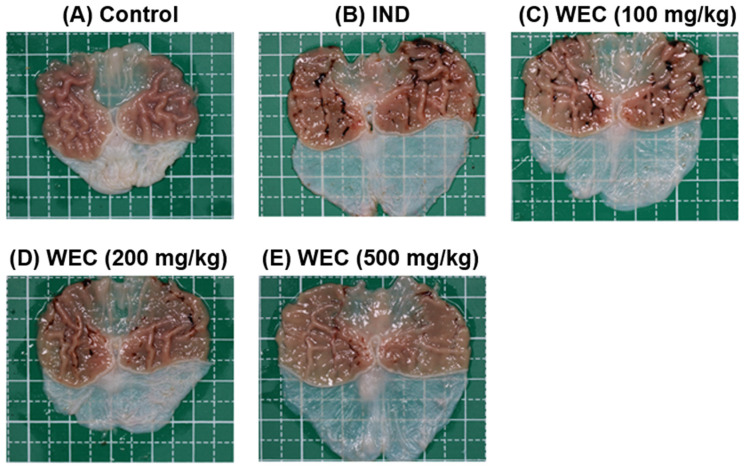
The gross appearance of gastric mucosa of control or IND-treated rats. Rats were administered indicated dose of WEC orally for 7 consecutive days. Then, 80 mg/kg of IND was given orally to rats on the last day of experiment and sacrificed 6 h later. The gastric tissue of rats was harvested and photographed. (**A**) Control group; (**B**) IND group, treated with IND (80 mg/kg); (**C**) Low dose WEC group, treated with WEC (100 mg/kg) + IND (80 mg/kg); (**D**) Medium dose WEC group, treated with WEC (200 mg/kg) + IND (80 mg/kg); (**E**) High dose WEC group, treated with WEC (500 mg/kg) + IND (80 mg/kg).

**Figure 4 foods-12-00156-f004:**
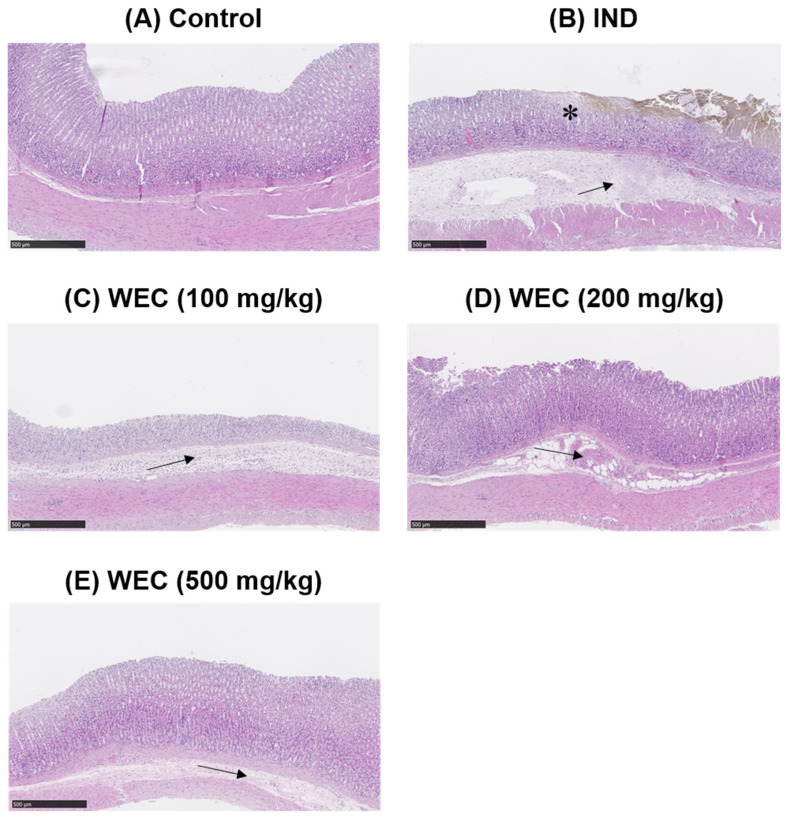
Histopathological assessment of gastric mucosa of control or IND-treated rats (H&E staining, scale bars = 500 μm). * indicates damaged mucosal epithelium with disrupted glandular structure. Arrow depicts edema and inflammatory cell infiltration in the submucosa. (**A**) Control group; (**B**) IND group, treated with IND (80 mg/kg); (**C**) Low dose WEC group, treated with WEC (100 mg/kg) + IND (80 mg/kg); (**D**) Medium dose WEC group, treated with WEC (200 mg/kg) + IND (80 mg/kg); (**E**) High dose WEC group, treated with WEC (500 mg/kg) + IND (80 mg/kg).

**Figure 5 foods-12-00156-f005:**
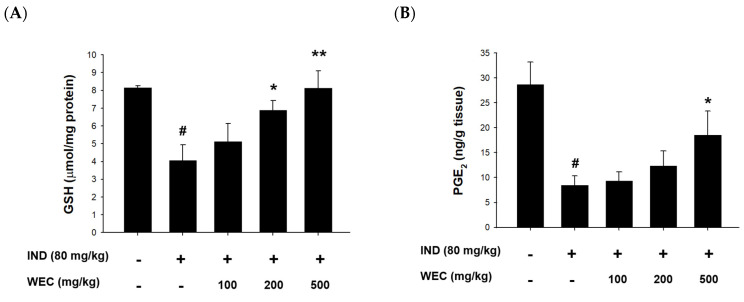
Effects of WEC on the contents of GSH and PGE_2_ and the protein levels of COX-1 and COX-2 of gastric mucosa in rats. The gastric contents of (**A**) GSH and (**B**) PGE_2_ were determined using commercial assay kits. Data are represented as means ± SD (n = 6). The protein expression of (**C**) COX-1 and (**D**) COX-2 was analyzed by Western blot. Data are represented as means ± SD (n = 3). # *p* < 0.05 compared with the control group. * *p* < 0.05, ** *p* < 0.01, *** *p* < 0.001 compared with the IND-treated group. + and − indicate with and without treatment, respectively.

**Figure 6 foods-12-00156-f006:**
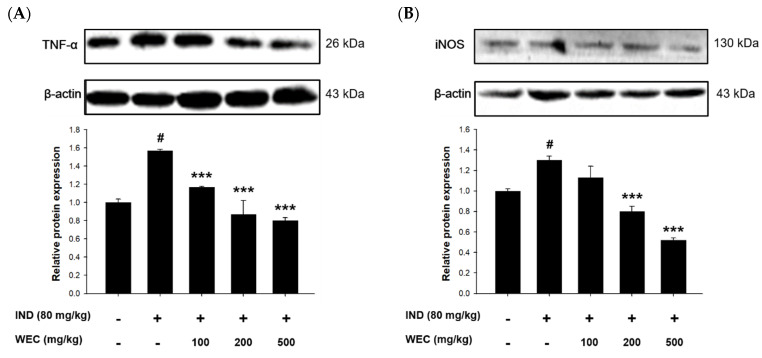
Effects of WEC on the protein levels of (**A**) TNF-α and (**B**) iNOS of gastric mucosa in rats analyzed by Western blot. Data are represented as means ± SD (n = 3). # *p* < 0.05 compared with the control group. *** *p* < 0.001 compared with the IND-treated group. + and − indicate with and without treatment, respectively.

**Table 1 foods-12-00156-t001:** Amino acid composition of WEC and its hydrolyzate.

	Free Amino Acid (mg/g)	Hydrolysate (mg/g)
Aspartic acid	0.41 ± 0.03	66 ± 4
Glutamic acid	0.59 ± 0.05	93 ± 9
Serine	0.85 ± 0.07	30 ± 2
Histidine	0.30 ± 0.02	9 ± 1
Glycine	0.27 ± 0.03	24 ± 2
Threonine	1.64 ± 0.07	26 ± 2
Arginine	2.1 ± 0.1	44 ± 3
Alanine	0.99 ± 0.08	30 ± 1
Tyrosine	0.97 ± 0.06	19 ± 1
Cysteine	n.d.	33 ± 3
Valine	3.0 ± 0.2	30 ± 1
Methionine	1.35 ± 0.08	10 ± 1
Phenylalanine	2.4 ± 0.2	30 ± 2
Isoleucine	3.4 ± 0.3	34 ± 2
Leucine	6.8 ± 0.5	50 ± 5
Lysine	2.13 ± 0.09	56 ± 4
Proline	0.25 ± 0.01	25 ± 1
Tryptophan	0.62 ± 0.03	n.d.
Asparagine	0.65 ± 0.04	n.d.
Taurine	0.24 ± 0.02	n.d.

The amino acid composition of WEC and its hydrolyzate were analyzed using an automatic amino acid analyzer. Data were expressed as mean ± SD of three determinations. n.d. indicates not detected.

**Table 2 foods-12-00156-t002:** Effects of WEC on gastric ulcer area and preventive index in rats treated with IND.

Group	Ulcer Area (mm^2^)	Preventive Index (%)
Control	0 ± 0	─
IND	75 ± 26 ^#^	─
WEC (100 mg/kg)	47 ± 14 *	38
WEC (200 mg/kg)	37 ± 9 **	51
WEC (500 mg/kg)	20 ± 6 ***	74

The ulcer area was quantified using Image J. Data are represented as means ± SD (n = 6). # *p* < 0.05 compared with the control group. * *p* < 0.05, ** *p* < 0.01, *** *p* < 0.001 compared with the IND-treated group. Preventive index (%) = [(Ulcer area_IND group_ − Ulcer area_WEC group_)/Ulcer area_IND group_] × 100%.

## Data Availability

Data are contained within the article.

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
