# Peer review of "Freshwater Clam Extract Attenuates Indomethacin-Induced Gastric Damage In Vitro and In Vivo"

_foods, 2022, doi:10.3390/foods12010156_

Round 1

Reviewer 1 Report

The authors have carried out a research work to evaluate the mechanisms of action by which an water extract of clam (WEC) can reduce the damage to the gastric mucosa produced by non-steroidal anti-inflammatory drugs (NSAIDs). Nowadays, the consumption of this type of medication is quite common, leading to its adverse effects, such as gastric damage and the consequent appearance of gastric ulcers. Therefore, it is of vital interest to find natural alternatives that can prevent this, and therefore reduce the incidence of this pathology. The authors use an appropriate in vitro and in vivo model according to the effects to be sought. The results obtained elucidate already described mechanisms of action of essential amino acids and WEC, but in a tissue not yet evaluated. Therefore, the work is novel and interesting as it provides new evidence that allows the use of this ingredient for gastric health in food supplements.

However, there are some mistakes that should be checked.

The term NSAIDs has been introduced at line 45, so there is no need to introduce it again on line 66.

It is mentioned twice in a row that prostaglandins have protective effects on the mucosa on lines 69 and 70.

Add a reference which demonstrates that indomethacin is the NSAID that causes the most damage to the gastric mucosa in rats (lines 72-73).

What part of the clam has been used to make the extract? It should be indicated where the clam powder comes from.

Indicate the extraction temperature to obtain the WEC (line 120).

Add the brand and the manufacturer (Brand; Manufacturer, Country) of the used equipment at the Material and Methods section: aminoacid analyzer (line 124), fluorescence spectrometer (140) and commercial assay kits for GSH and PGE2 quantification (line 176).

Indicate the manufacturer and the used dilutions of the primary antibodies used at the Western blot analysis (line 186).

Sometimes the formatting of the text does not follow the general format (for example, lines 168 and 187).

I highly recommend reviewing the correct use of English in the document. There are sentences that are not well formulated, such as “These essential amino acids also exert a considerable content in the WEC 210 hydrolysate, while the content of Glu in the hydrolyzed WEC is the highest” in line 210-211.

Indicate at the Table 1 footer the meaning of “n.d.”. Also add this indication at the non-detected aminoacids of the hydrolysate.

If there are statistical differences between the “Control group” and the “IND + WEC group” at the different Figures and Tables, please, add the respective indicators (#). It is important to know when the prevention of IND-induced damage is like that of a control situation.

Please, use the correct terms referred to “protein levels”. For example, change “protein expression” for “protein levels” at lines 265, 268 and on Figure 2. Western blot technique measures protein levels. To measure the “expression” the levels of their mRNA should be quantified.

It is stated on line 400 that nitric oxide (NO) is a cytokine. Nitric oxide is not a cytokine but a signalling molecule. Please correct “(…) to the production of NO and pro-inflammatory cytokines such as IL-6 and TNF-α.

Some figure captions can be simplified. For example, the one in Figure 6 to “Effects of WEC on the protein levels of (A) TNF-α and (B) iNOS of gastric mucosa in rats analyzed by western blot”.

Minor mistakes:

Change “(…) IND treatment, pre-treated with (…)” to “(…) IND treatment, and pre-treatment with (…) on line 23

Change “Earlier investigates” to “Previous studies” on line 76

Change “(…) South Asia, widely (…)” to “(South Asia, is widely (…)”

“emperature” on line 144

Add the correct degree value on line 145: “2e ± 2ºC”

Change “(…) and the rats were sacrificed (…)” to “(…) and sacrificed using (…)” on lines 153-154

Change “modified” to “modifications” on line 168

Change “incubated” to “incubations” on line 185

“of f amino” on line 206

Change “(…) Val, have higher amount.” to “Val, stand out.”.

Change “harvested” to “collected” on line 289.

Change “The histopathological assessment (…)” to “Histopathological assessment (…)” on line 331.

Author Response

Comments:

The authors have carried out a research work to evaluate the mechanisms of action by which an water extract of clam (WEC) can reduce the damage to the gastric mucosa produced by non-steroidal anti-inflammatory drugs (NSAIDs). Nowadays, the consumption of this type of medication is quite common, leading to its adverse effects, such as gastric damage and the consequent appearance of gastric ulcers. Therefore, it is of vital interest to find natural alternatives that can prevent this, and therefore reduce the incidence of this pathology. The authors use an appropriate in vitro and in vivo model according to the effects to be sought. The results obtained elucidate already described mechanisms of action of essential amino acids and WEC, but in a tissue not yet evaluated. Therefore, the work is novel and interesting as it provides new evidence that allows the use of this ingredient for gastric health in food supplements.

However, there are some mistakes that should be checked.

  1. The term NSAIDs has been introduced at line 45, so there is no need to introduce it again on line 66.

Response: Thanks for reviewer's suggestion. We have corrected it (line 68).

  1. It is mentioned twice in a row that prostaglandins have protective effects on the mucosa on lines 69 and 70.

Response: Thanks for reviewer's suggestion. We have revised it (line 71).

  1. Add a reference which demonstrates that indomethacin is the NSAID that causes the most damage to the gastric mucosa in rats (lines 72-73).

Response: Thanks for reviewer's suggestion. A reference has been added to this sentence (ref. 16).

  1. What part of the clam has been used to make the extract? It should be indicated where the clam powder comes from.

Response: Thanks for reviewer's suggestion. We have addressed that the dried powder made from freshwater clam meat was kindly provided by Li Chuan Aquafarm Co., Ltd. (Hualien, Taiwan) in the Materials and Methods section (lines 118-119).

  1. Indicate the extraction temperature to obtain the WEC (line 120).

Response: Thanks for reviewer's suggestion. We have added the extraction temperature (at room temperature) to obtain the WEC (line 122).

  1. Add the brand and the manufacturer (Brand; Manufacturer, Country) of the used equipment at the Material and Methods section: amino acid analyzer (line 124), fluorescence spectrometer (140) and commercial assay kits for GSH and PGE2 quantification (line 176).

Response: Thanks for reviewer's suggestion. We have added the manufacturer (Brand, Manufacturer, Country) of the used equipment in the Material and Methods section: amino acid analyzer (L-8900 AAA, Hitachi High Tech Analytical Science Ltd., Japan) (lines 125-126), fluorescence spectrometer (Hitachi F-2700, Hitachi High Tech Analytical Science Ltd., Japan) (line 145) and commercial assay kits for GSH and PGE2 quantification (Elabscience Biotechnology, Houston, TX, USA) (lines 181-182).

  1. Indicate the manufacturer and the used dilutions of the primary antibodies used at the Western blot analysis (line 186).

Response: Thanks for reviewer's suggestion. The manufacturer of the antibodies has been described in the 2.1. Chemical Reagents in the Materials and Methods section (lines 103-105). The used dilution of the primary antibodies has been addressed in 2.8. Protein Expression Analysis (1:1000) (line 191).

  1. Sometimes the formatting of the text does not follow the general format (for example, lines 168 and 187).

Response: Thanks for reviewer's reminder. We have revised the format of these two sentences (lines 173 and 192).

  1. I highly recommend reviewing the correct use of English in the document. There are sentences that are not well formulated, such as “These essential amino acids also exert a considerable content in the WEC 210 hydrolysate, while the content of Glu in the hydrolyzed WEC is the highest” in lines 210-211.

Response: Thanks for reviewer's suggestion. We have reworded the sentence as “Although the Glu was the highest in the WEC hydrolyzate, those essential amino acids measured in the free amino acids also had a certain content in the WEC hydrolysate” (lines 215-216).

  1. Indicate at the Table 1 footer the meaning of “n.d.”. Also add this indication at the non-detected aminoacids of the hydrolysate.

Response: Thanks for reviewer's suggestion. We have added the Table 1 footer the meaning of “n.d.” and this indication at the non-detected amino acids of the hydrolysate.

  1. If there are statistical differences between the “Control group” and the “IND + WEC group” at the different Figures and Tables, please, add the respective indicators (#). It is important to know when the prevention of IND-induced damage is like that of a control situation.

Response: Thanks for reviewer's suggestion. Thank you for the suggestion, but in order to avoid confusion caused by too many symbols on the bars, most of the literature marked these symbols to indicate the statistical differences like our results. We hope you will kindly forgive such a representation.

  1. Please, use the correct terms referred to “protein levels”. For example, change “protein expression” for “protein levels” at lines 265, 268 and on Figure 2. Western blot technique measures protein levels. To measure the “expression” the levels of their mRNA should be quantified.

Response: Thanks for reviewer's suggestion. We have changed “protein expression” for “protein levels” at lines 265, 268 and on Figures 2, 5 and 6 (lines 255, 270, 272, 282, 342, 379, 385, and 414).

  1. It is stated on line 400 that nitric oxide (NO) is a cytokine. Nitric oxide is not a cytokine but a signalling molecule. Please correct “(…) to the production of NO and pro-inflammatory cytokines such as IL-6 and TNF-α.

Response: Thanks for reviewer's suggestion. We have corrected it (line 407).

  1. Some figure captions can be simplified. For example, the one in Figure 6 to “Effects of WEC on the protein levels of (A) TNF-α and (B) iNOS of gastric mucosa in rats analyzed by western blot”.

Response: Thanks for reviewer's suggestion. We have simplified the figure caption of Figure 6 as your suggestion (lines 414-415).

  1. Minor mistakes.

Response: We would like to thank the reviewers for the minor mistakes in our manuscript, and we have corrected these minor mistakes (lines 23, 77, 87, 149, 150, 158-159, 173, 190, 211, 214-215, 294, and 339).

Reviewer 2 Report

The authors describe a convincing evidence that a water extract from freshwater clam provides a partial protection against gastric damage induced by indomethacin both in vitro in rat gastric mucosa RGM-1 cells and in vivo in rats.  The experiments are well performed and described.

·        The authors propose, in order to explain the beneficial effect of WEC, that this could result from an anti-inflammatory activity of amino-acids and/or oligopeptides present in the water extract.  Accordingly, a control including only the amino acids whose composition in WEC has been identified, should have been included to further evidence their role in the effects of WEC on the digestive tract cells.  The same for a possible effect of the fatty acids in freshwater clam on the inhibition the NF-kB pathway.  It is indeed crucial to evidence whether the WEC extract provides an additional or even synergistic advantage on a simple mixture of amino acids, supplemented or not with some fatty acids.

·        The authors mention that the presence of high amounts of glutamic acid in WEC could be crucial to explain the effects that were observed.  Nevertheless, if it is possible, they should remember that, from a biochemical viewpoint, this amino acid is one of the easiest one for cells to produce from different substrates.  Here also, a control with glutamic acid should validate their hypothesis.

·        The authors show the amino acids composition of WEC, but do not really provide information on the other constituents, except to mention the presence of “phenolic compounds and steroid components”.  This should be clarified.

·        It seems clear that this study did not aim at developing a new treatment for rat gastric ulcer…, but, more logically, to provide new tracks for the human disease.  They indeed claim that “[WEC] can be developed as a relevant nutritional supplement in the future”.  Therefore, the authors should discuss the possibility and feasibility of using the human equivalent of 500 mg/kg in rats, i.e. ± 81 mg/kg, thus from ± 4 to 8 g per day for “mean” people.

·        The abstract should indicate that the authors propose that the beneficial effects of WEC could result from the presence of essential amino acids and oligopeptides through an anti-inflammatory potential.

Author Response

Comments:

The authors describe a convincing evidence that a water extract from freshwater clam provides a partial protection against gastric damage induced by indomethacin both in vitro in rat gastric mucosa RGM-1 cells and in vivo in rats. The experiments are well performed and described.

  1. The authors propose, in order to explain the beneficial effect of WEC, that this could result from an anti-inflammatory activity of amino-acids and/or oligopeptides present in the water extract. Accordingly, a control including only the amino acids whose composition in WEC has been identified, should have been included to further evidence their role in the effects of WEC on the digestive tract cells.  The same for a possible effect of the fatty acids in freshwater clam on the inhibition the NF-kB pathway.  It is indeed crucial to evidence whether the WEC extract provides an additional or even synergistic advantage on a simple mixture of amino acids, supplemented or not with some fatty acids.

Response: Thanks for reviewer's suggestion. We understand that a control including only the amino acids whose composition in WEC has been identified you mentioned is important for further evidence their role in the effects of WEC on the digestive tract cells. Therefore, we will investigate this aspect and also consider the protective effect of fatty acids in the future research work.

  1. The authors mention that the presence of high amounts of glutamic acid in WEC could be crucial to explain the effects that were observed. Nevertheless, if it is possible, they should remember that, from a biochemical viewpoint, this amino acid is one of the easiest one for cells to produce from different substrates.  Here also, a control with glutamic acid should validate their hypothesis.

Response: Thanks for reviewer's reminder. We know that Glu is one of the easiest one for cells to produce from different substrates from a biochemical viewpoint. However, several studies have proved or mentioned that supplementing Glu in the diet is beneficial to the growth of organisms or the synthesis of glutathione, such as “Broiler responses to reduced-protein diets supplemented with valine, isoleucine, glycine, and glutamic acid” published by Berres et al. in 2010 and “Impact of supplementary amino acids, micronutrients, and overall eiet on glutathione homeostasis” published by Gould et al. in 2019. A control with glutamic acid will be considered to be investigated in our future research work.

  1. The authors show the amino acids composition of WEC, but do not really provide information on the other constituents, except to mention the presence of “phenolic compounds and steroid components”. This should be clarified.

Response: Thanks for reviewer's reminder. Indeed, we believe that WEC might possess other active components such as phenolic compounds and steroid components (the content of steroid should be very low in the aqueous extract). However, this study focused on whether the protective effect of WEC on gastric mucosa is related to its amino acid composition. We will also consider to conduct related research for other active components in the future.

  1. It seems clear that this study did not aim at developing a new treatment for rat gastric ulcer…, but, more logically, to provide new tracks for the human disease. They indeed claim that “[WEC] can be developed as a relevant nutritional supplement in the future”.  Therefore, the authors should discuss the possibility and feasibility of using the human equivalent of 500 mg/kg in rats, i.e. ± 81 mg/kg, thus from ± 4 to 8 g per day for “mean” people.

Response: Thanks for reviewer's suggestion. The discussion for the possibility and feasibility of using the human equivalent of 100-500 mg/kg in rats was addressed in the section 3.4. (lines 321-324)

  1. The abstract should indicate that the authors propose that the beneficial effects of WEC could result from the presence of essential amino acids and oligopeptides through an anti-inflammatory potential.

Response: Thanks for reviewer's suggestion. We have added the beneficial effects of WEC could result from the presence of essential amino acids and oligopeptides through an anti-inflammatory potential to Abstract section (lines 33-34).

Reviewer 3 Report

The authors reported how Freshwater Clam Extract can attenuates Indomethacin-induced gastric damage In Vitro and In Vivo

Authors need to address some comments

1- Why did the authors choose the described treatment duration (2 hrs and 6 hrs)

2- 24 hr treatment needs to be added for the in vitro experiments

3- ROS detection with DCF staining need to be re-written

4- Please provide the original Blot for the WB detection

Author Response

Comments:

The authors reported how Freshwater Clam Extract can attenuates Indomethacin-induced gastric damage In Vitro and In Vivo

Authors need to address some comments

  1. Why did the authors choose the described treatment duration (2 hrs and 6 hrs)?

Response: Thanks for reviewer's reminder. We refer to the method in the literature published by Furukawa and Okabe in 1997 and the results from our preliminary experiments. We believe that the treatment duration we used should show a reasonable trend. We have added the citation of this article to section of 2.3. (lines 131-132). Furthermore, most of the food stay in the stomach for 2-3 h in theory, so the sample treatment duration we used should be reasonable.

  1. 24 hr treatment needs to be added for the in vitro experiments

Response: Thanks for reviewer's suggestion. As we mentioned in our reply to your first comment, we refer to the method in the literature published by Furukawa and Okabe in 1997 and the results from our preliminary experiments. We believe that the treatment duration we used should show a reasonable trend. In the future, we will consider to use 24 h treatment you suggested to evaluate whether the sample has a better protective effect on gastric mucosa cells. Thank you a lot for this suggestion.

  1. ROS detection with DCF staining need to be re-written

Response: Thanks for reviewer's suggestion. We have rewritten the method of ROS detection with DCF staining (lines 136-146).

  1. Please provide the original Blot for the WB detection.

Response: Thanks for reviewer's suggestion. We have provided the original blot image for the WB detection. Please see the supplement.

Round 2

Reviewer 2 Report

The authors take into consideration my remarks… but defer the required controls… to future work!

 In particular, they state:  We understand that a control including only the amino acids whose composition in WEC has been identified you mentioned is important for further evidence their role in the effects of WEC on the digestive tract cells. Therefore, we will investigate this aspect and also consider the protective effect of fatty acids in the future research work.”…

… and later: “Indeed, we believe that WEC might possess other active components such as phenolic compounds and steroid components (the content of steroid should be very low in the aqueous extract). However, this study focused on whether the protective effect of WEC on gastric mucosa is related to its amino acid composition. We will also consider to conduct related research for other active components in the future.”

 How can the authors reconcile their statement that the protective effect they observed on the gastric mucosa is related to the presence of amino acids, and not include a control in the presence of the amino acids and in the absence of the other compounds of the clam extract?

I still consider that this work is interesting, but, according to me, the appropriate controls are mandatory.  In its present form, not in the future!

Author Response

Reviewer: 2

Comments:

The authors take into consideration my remarks… but defer the required controls… to future work!

In particular, they state:  “We understand that a control including only the amino acids whose composition in WEC has been identified you mentioned is important for further evidence their role in the effects of WEC on the digestive tract cells. Therefore, we will investigate this aspect and also consider the protective effect of fatty acids in the future research work.”…

… and later: “Indeed, we believe that WEC might possess other active components such as phenolic compounds and steroid components (the content of steroid should be very low in the aqueous extract). However, this study focused on whether the protective effect of WEC on gastric mucosa is related to its amino acid composition. We will also consider to conduct related research for other active components in the future.”

How can the authors reconcile their statement that the protective effect they observed on the gastric mucosa is related to the presence of amino acids, and not include a control in the presence of the amino acids and in the absence of the other compounds of the clam extract?

I still consider that this work is interesting, but, according to me, the appropriate controls are mandatory.  In its present form, not in the future!

Response: Thanks a lot for reviewer's suggestion. We understand that the appropriate controls might be meaningful to clarify whether the protective ability of WEC is from amino acids. However, our research focused on whether WEC has the effect of protecting the gastric mucosa and whether it has the potential to be developed as a “health food” or “nutritional supplement”. While observing that WEC has the effect of protecting the gastric mucosa, we also found that it has a higher content of amino acids. Moreover, according to previous literature, amino acids have protective effects on gastric mucosa (reference number 35 in our manuscript). Therefore, we speculated that at least part of the protective effect of WEC on gastric mucosa is due to the amino acids it contains. We believe this should be a reasonable inference in “foods” research. Of course, in the future, we will also conduct relevant research according to the reviewer's suggestion to clarify the issues he mentioned. Thanks again to the reviewers for their meaningful suggestions for our study.

Reviewer 3 Report

Thanks for the Responses

please consider better blocking and Antibodies for WB in the future studies

Author Response

Reviewer: 3

Comments:

Thanks for the Responses

Please consider better blocking and Antibodies for WB in the future studies

Response: Thanks for reviewer's suggestion. We will use better blocking and antibodies for WB in the future studies.